# Novel genes involved in the genetic architecture of temperament in Brahman cattle

Francisco Alejandro Paredes-Sánchez[1], Ana María Sifuentes-Rincón[1]*,
Eduardo Casas[2], Williams Arellano-Vera[1], G. Manuel Parra-Bracamonte[1], David G. Riley[3],
Thomas H. Welsh Jr.[3], Ronald D. Randel[4]

**1** CBG, Instituto Politécnico Nacional, Reynosa, Tamaulipas, México, **2** USDA, ARS, National Animal Disease Center, Ames, IA, United States of America, **3** Texas A&M University, College Station, TX, United States of America, **4** Texas A&M AgriLife Research, Overton, TX, United States of America

\* asifuentes@ipn.mx

**Data Availability Statement:** All relevant data are within the paper and its Supporting Information files.

## Abstract

Cattle temperament is a complex and economically relevant trait. The objective of this study was to identify genomic regions and genes associated with cattle temperament. From a Brahman cattle population of 1,370 animals evaluated for temperament traits (Exit velocity-EV, Pen Score-PS, Temperament Score-TS), two groups of temperament-contrasting animals were identified based on their EV-average values ±1/2 standard deviation (SD). To be considered in the calm group, the EV of females ranged between 0.16–1.82 m/s (n = 50) and the EV of males ranged between 0.4–1.56 m/s (n = 48). Females were classified as temperamental if their EV ranged between 3.13–7.66 m/s (n = 46) and males were classified as temperamental if their EV ranged between 3.05–10.83 m/s (n = 45). Selected animals were genotyped using a total of 139,376 SNPs (GGP-HD-150K), evaluated for their association with EV. The Genome-Wide Association analysis (GWAS) identified fourteen SNPs: rs135340276, rs134895560, rs110190635, rs42949831, rs135982573, rs109393235, rs109531929, rs135087545, rs41839733, rs42486577, rs136661522, rs110882543, rs110864071, rs109722627, (*P*<8.1E-05), nine of them were located on intergenic regions, harboring seventeen genes, of which only *ACER3*, *VRK2*, *FANCL* and *SLCO3A1* were considered candidate associated with bovine temperament due to their reported biological functions. Five SNPs were located at introns of the *NRXN3*, *EXOC4*, *CACNG4* and *SLC9A4* genes. The indicated candidate genes are implicated in a wide range of behavioural phenotypes and complex cognitive functions. The association of the fourteen SNPs on bovine temperament traits (EV, PS and TS) was evaluated; all these SNPs were significant for EV; only some were associated with PS and TS. Fourteen SNPs were associated with EV which allowed the identification of twenty-one candidate genes for Brahman temperament. From a functional point of view, the five intronic SNPs identified in this study, are candidates to address control of bovine temperament, further investigation will probe their role in expression of this trait.

**Funding:** AMSR received support by Consejo Nacional de Ciencia y Tecnología (https://www. conacyt.gob.mx) through the project CONACyT 294826 and 299055 CONACyT and from the Instituto Politécnico Nacional (http://www.sappi. ipn.mx/) through the research Project SIP 20195072. RR,TW and RR by Texas A&M AgriLife Research. The funders had no role in study design, data collection and analysis, decision to publish, or preparation of the manuscript.

**Competing interests:** The authors have declared that no competing interests exist.

## Introduction

Cattle temperament is described as the animal's response to handling [1, 2]. Animals with extreme temperaments exhibit a wide variation in production traits; more excitable, temperamental animals have negative effects reported for weight gain, reproductive efficiency, milk production, meat quality and higher disease susceptibility while docile animals, had positive effects [3]. For example, weight gain of docile cattle has been reported to be 10–14% higher than that observed for temperamental animals [3].

Genetic background plays an important role in expression of temperament. For example, Hohenboken [4] found that Brahman (*Bos indicus*) cattle behave differently in corrals and working facilities compared to *Bos taurus* cattle. Recent studies point out that temperament of *Bos indicus*, purebred or Brahman crossbred cattle, negatively impacts performance, human safety and animal welfare, thus their inclusion and use sometimes is avoided, missing the benefits of their superior adaptability to tropical climatic conditions [5]. Efforts to improve the temperament in Brahman cattle have included the indirect estimation of heritability ($h^2$) of this trait using exit velocity (EV), which was estimated in $h^2 = 0.27 \pm 0.1$ [6], and the use of different approaches to determine the molecular basis of this trait in *Bos indicus*. Hulsman et al. [7] studied the association of 54,609 Single Nucleotide Polymorphisms (SNPs) with bovine temperament in a population of Nellore-Angus cattle, measuring it by overall temperament at weaning, based on social separation. Hulsman et al. [7] identified 37 genomic regions associated with temperament and located 172 genes, and after an enrichment analysis, identified significant terms of gene ontology, related to sodium ion transport. Valente et al. [8] used flight speed to measure bovine temperament in a Nellore population, identifying 6 new candidate genes. In a study of a Guzerat population, dos Santos et al. [9] used the BovineSNP50 v2 array and measured the reactivity (the frequency and intensity of its movements while the animal is confined) as a temperament test, in which seven candidate genes related to temperament were identified. These reports not only indicate that genetic control of cattle temperament involves a wide network of genes but also that the breeds and the tests applied to investigate this trait are fundamental to the results obtained. Given the relevance of the Brahman breed in the context of their genetic predisposition to have more excitable temperaments, the objective of this study was to identify genes affecting temperament in two extreme groups, i.e. calm and temperamental Brahman animals selected according to their temperament phenotypes defined by their EV. The candidate genes identified in this work may impact not only basic research but also the application of genomic tools for the improvement of cattle. Temperament is economically relevant for beef cattle production as those with a calm temperament that have better growth performance than more temperamental animals. Identification of markers that allow detection of temperamental animals early in life (e.g., newborn animals) would help Brahman farmers in decision making and will improve human safety and animal welfare.

## Materials and methods

### Ethical statement

All procedures were in compliance with the Guide for the Care and Use of Agricultural Animals in Research and Teaching 2010 and approved by the Texas A&M University Animal Care and Use Committee AUP 2002–315.

### Animals and collected samples

One hundred and eighty-nine animals were selected from a population of 1,370 Brahman calves (628 male and 742 female) born between 2002 and 2017 at the Texas A&M AgriLife

Research and Extension Center in Overton, TX. The management of the Brahman cattle resource herd and data recording were previously reported by [6]. All temperament data:

- Exit Velocity (EV), an objective measure, the rate at which the animals exited the working chute and traveled a distance of 1.83m, measured with an infrared sensor (FarmTek Inc., North Wylie, TX, USA) [10, 11].

- Pen Score (PS), a subjective measure, based on individual visual assessments of animal behavior while confined to a pen in groups of five animals, where a score of 1 is calm and 5 aggressive [12]; and,

- Temperament Score (TS) were calculated by averaging the PS and EV [TS = (PS + EV)/2] [13], determined at the time that the calves were weaned from their dams.

Ear notches obtained at the time of weaning were snap frozen and were stored at -80°C until used to determine genotypes.

In this study a selective genotyping approach was applied, this strategy has been used in the analysis of other traits in cattle to facilitate the detection of causative alleles, due to an enrichment of these alleles among phenotypically extreme individuals [14]. Unlike other temperament evaluations, the EV and PS have been shown to be positively correlated (r ≥ 0.35, $P$ <0.005) (11). On the basis of EV heritability calculated in the Brahman population used in this work ($h^2$ = 0.27) [6] and considering EV as an objective measure that offers higher reliability between evaluators than subjective methods (PS), the selective genotyping was achieved following the strategy of tailed EV deviations; the animals were classified into calm and temperamental, based on the sex-group average values (EV) ± 1/2 standard deviation (SD). Of the total population, we identified two groups of temperament-contrasting animals, the calmest and most temperamental. Females were calm if their EV was within the range of 0.16–1.82 m/s (n = 50) and males were calm if their EV was within the range of 0.4–1.56 m/s (n = 48). Females were temperamental if their EV was within the range of 3.13–7.66 m/s (n = 46) and males were temperamental if their EV was within the range of 3.05–10.83 m/s (n = 45).

## Genotyping and GWAs

The DNA extraction from ear notch samples and genotyping of the 189 animals were done using the GeneSeek Genomic Profiler HD 150K chip (GGP-HD-150K) (Neogen, Lincoln, NE) by the NEOGEN company. The GWAS analyses were performed with PLINK 1.9 software [15] using the case-control method for unrelated animals, considering calm and temperamental animals as case and control, respectively. The algorithm considers a single marker association analyses through frequency comparison using a chi-square test. As part of quality control, the following SNP were excluded: a) those with an unknown genomic position; b) those located on a sex chromosome; c) monomorphic SNPs; d) SNPs with minor allele frequency (MAF) below 0.01, and, a call rate below 90%. Individuals with more than 10% missing genotypes were also excluded. After quality control, 172 genotyped animals [85 males and 87 females; 89 calm (46% males; 54% females) and 83 temperamental (53% males; 47% females)] were used. SNPs with $P$-values $< 8 \times 10{-}5$ were considered significant and associated with EV, and therefore to bovine temperament. From the coordinates provided by the chip information, the physical position of significant SNPs was obtained based on the last available *Bos taurus* genome (ARS-UCD 1.2) using the Genome Data Viewer software available at the National Center for Biotechnology Information (NCBI). Candidate genes were selected based on their location in relation with the significant SNPs, through the genome navigator. All genes around 300Kb downstream and upstream, were selected and analysed according to literature reports.

### Effects of significant SNPs with temperament traits

To assess the effect of significant SNPs on EV, PS and TS, a mixed model was fitted including the fixed effect of sex of calf, individual effect of fourteen loci (rs135340276, rs134895560, rs110190635, rs42949831, rs135982573, rs109393235, rs109531929, rs135087545, rs41839733, rs42486577, rs136661522, rs110882543, rs110864071, and rs109722627). Year of birth was considered as random effect, to capture and remove actual trend effects of this factor on response variables and avoid the use of unnecessary degrees of freedom. The model was fitted using the MIXED procedure. Genotypes with frequency lower than 0.05 were excluded from the analysis. Least square means of the genotypes were estimated and compared by a *t*-test with a Bonferroni adjustment using the PDIFF statement. All statistical analyses were performed using SAS 9.0 (SAS Inst. Inc., Cary, NC, USA).

## Results

A total of 104,235 SNPs were evaluated for their association with EV in Brahman cattle. On average, 4,132 SNPs were evaluated in each BTA (*Bos taurus* autosome). The BTA 1 and 25 exhibited the highest and lowest numbers of SNP, respectively. The average distance between adjacent SNPs was 21,039 bp, and the minimum (2 bp) and maximum distances (3,882,807 bp) between adjacent SNP were found on BTA 18 and 5, respectively. The Fig 1 depicts a Manhattan plot with results from the genomic analysis.

The continuous line represents the threshold considered ($P < 8.1E{-}05$). According to the significance threshold considered ($P{<}8.1E{-}05$), fourteen SNPs were associated with EV (Table 1). Nine were located on intergenic regions and five on introns of genes *NRXN3* (neurexin 3; GeneID: 614412) *EXOC4* (exocyst complex component 4; GenID: 537690), *CACNG4* (calcium voltage-gated channel auxiliary subunit gamma 4; GeneID: 519331) and *SLC9A4* (solute carrier family 9 member A4; GenID: 536970).

Seventeen genes were mapped around the significant SNPs located on intergenic regions (Table 2). These genes were *LOC10713318* (uncharacterized), *TSKU* (tsukushi, small leucine rich proteoglycan), *LOC112441553* (uncharacterized), *GUCY2E* (guanylate cyclase 2E), *LOC112441534* (uncharacterized), *LRRC32* (leucine rich repeat containing 32), *LOC1071 33180* (uncharacterized), *EMSY* (EMSY transcriptional repressor, BRCA2 interacting), *ACER3* (alkaline ceramidase 3) on BTA 15, while the genes *VRK2* (VRK serine/threonine kinase 2), *FANCL* (FA complementation group L) and *TRNAC-ACA* (transfer RNA cysteine), were located on BTA 11. The genes *SLCO3A1* (solute carrier organic anion transporter family member 3A1) and *LOC112443232* (uncharacterized) were located on BTA 21. Finally, the genes

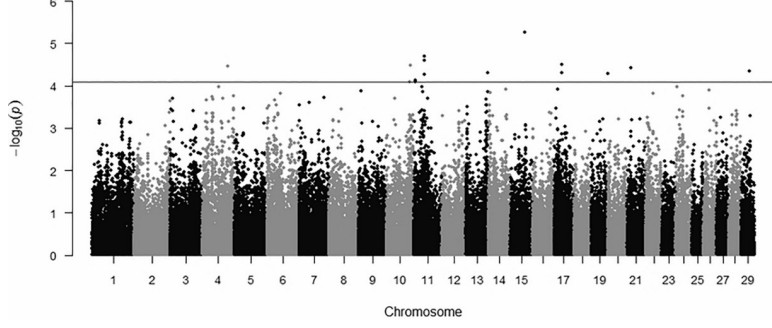

**Fig 1. Manhattan plot of the -log10 (*P*-values) for genomic association with temperament evaluated as EV in Brahman cattle.**

**Table 1. SNPs associated with temperament evaluated as EV in Brahman cattle.**

| SNP rs ID | BTA | Position (Mb) | Allele | MAF | Region | *P*-value |
|---|---|---|---|---|---|---|
| rs135340276 | 15 | 56.1 | [C/T] | 0.691/0.308 | Intergenic | 5.46E-06 |
| rs134895560 | 11 | 41.0 | [A/G] | 0.634/0.365 | Intergenic | 2.03E-05 |
| rs110190635 | 11 | 40.9 | [A/G] | 0.443/0.556 | Intergenic | 2.47E-05 |
| rs42949831 | 17 | 30.4 | [C/T] | 0.582/0.417 | Intergenic | 3.19E-05 |
| rs135982573 | 10 | 90.6 | [A/G] | 0.263/0.737 | Intron | 3.24E-05 |
| rs109393235 | 4 | 97.0 | [G/A] | 0.106/0.894 | Intron | 3.40E-05 |
| rs109531929 | 21 | 15.2 | [A/G] | 0.355/0.644 | Intergenic | 3.83E-05 |
| rs135087545 | 29 | 30.9 | [A/G] | 0.326/0.673 | Intergenic | 4.48E-05 |
| rs41839733 | 17 | 30.4 | [C/T] | 0.461/0.538 | Intergenic | 4.96E-05 |
| rs42486577 | 13 | 82.2 | [C/T] | 0.836/0.163 | Intergenic | 4.96E-05 |
| rs136661522 | 19 | 63.0 | [G/A] | 0.249/0.751 | Intron | 5.16E-05 |
| rs110882543 | 11 | 40.9 | [A/G] | 0.682/0.317 | Intergenic | 5.45E-05 |
| rs110864071 | 11 | 7.2 | [C/T] | 0.193/0.807 | Intron | 7.51E-05 |
| rs109722627 | 11 | 7.2 | [T/G] | 0.473/0.527 | Intron | 8.18E-05 |

BTA = *Bos taurus* autosome, Mb = Megabases, MAF = Minor allele frequency.

*LOC112449388* (uncharacterized), *LOC101903665* (uncharacterized) and *DOK5* (docking protein 5) were located on BTA 13. The genes *VRK2*, *FANCL* and *TRNAC-ACA* were identified by three SNPs associated with temperament (rs134895560, rs110190635 and rs11088254). For rs135087545, rs42949831 and rs41839733, no neighboring gene was identified.

## Association of significant SNPs

The association of the fourteen SNPs on three temperament traits, i.e., EV, PS and TS was determined. As shown in Table 3, all of the SNPs evaluated were associated with EV (*P*<0.01) only rs42486577 was associated with PS. The rs42949831, rs135982573, rs41839733 and rs110882543, were associated with TS. The most significant association with EV was shown for rs135087545, rs110864071 and rs109722627, located in the *SLC9A4*. The three SNPs are in LD ($r^2$ = 0.266), and, it should be noted that the SNP in the *EXOC4* gene had the greater significant difference between genotypic means for EV, with the carriers of GG genotype having EV values 2.64 m/s greater than the AA genotype (Table 3), while the homozygote GG genotype exhibited a TS that was 1.57 SD greater than those of the AA genotypes.

## Discussion

Currently, information about the molecular basis of temperament traits is scarce. Notably, even though the *Bos indicus* breeds seem to show the widest spectrum of temperament expression in reported studies, their use as a model is rare [16, 8]. Here we studied a Brahman population to identify candidate genes and genomic regions associated with cattle temperament, expressed as EV.

### Significant intergenic SNPs

The mapping of the regions around the intergenic SNPs identified in the GWAS allowed identification of seventeen candidate genes. Some of them such as *ACER3* (BTA 15), *VRK2* (BTA 11), *FANCL* (BTA 11), and *SLCO3A1* (BTA 21) could be considered to be candidate genes, due their reported biological functions. These genes were located on BTA that according to the

**Table 2. Genes mapped close (~300 Kb) to SNPs associated with EV of Brahman cattle.**

| SNP | Gene around 300 Kb | Distance | Description |
|---|---|---|---|
| | | Kb | |
| | LOC107133182 | 5.8 | Uncharacterized |
| rs135340276 | TSKU | 11 | Anterior commissure morphogenesis (GO:0021960) |
| | LOC112441553 | 52 | Uncharacterized |
| | GUCY2E | 85 | Guanylate cyclase activity (GO:0004383) |
| | LOC112441534 | 67.8 | Uncharacterized |
| | LRRC32 | 110.9 | Nucleoplasm (GO:0005654) |
| | LOC107133180 | 124 | Uncharacterized |
| | EMSY | 233 | Nucleoplasm (GO:0005654) |
| | ACER3 | 78.2 | Ceramidase activity (GO:0102121) |
| rs134895560 | VRK2 | 219.5 | Protein serine/threonine kinase activity (GO:0004674) |
| rs110190635 | FANCL | 131 | Ubiquitin protein ligase activity (GO:0061630) |
| rs110882543 | TRNAC-ACA | 87.7 | Uncharacterized |
| rs109531929 | SLCO3A1 | 57.2 | Sodium-independent organic anion transport (GO:0043252) |
| | LOC112443232 | 129.8 | Uncharacterized |
| rs42486577 | LOC112449388 | 50 | Uncharacterized |
| | LOC101903665 | 55.3 | Uncharacterized |
| | DOK5 | 220 | GTPase activator activity (GO:0005096) |

Kb = kilobases, GO = gene ontology.

Cattle Quantitative Trait Locus (QTL) Database (https://www.animalgenome.org/cgi-bin/QTLdb) [17], harbour QTLs related to bovine temperament: a) BTA 11 (Social separation—Standing alert; tendency of an animal to stand alert upon separation from its pen mates); b), BTA 15 (Temperament; milking temperament, pen score and reactivity).

The ACER3 gene catalyzes the hydrolysis of C18:1, the major unsaturated long-chain ceramide in the brain to sphingosine, its phosphorylated form sphingosine-1-phosphate (S1P), which has been implicated in the survival of neurons, and its dyshomeostasis has been associated with different neurodegenerative disorders [18]. It has been demonstrated in mice that ACER3 plays an important protective role, since it controls the homeostasis of ceramides and their derivatives such as S1P, avoiding the appearance of neurological disorders such as cerebellar ataxia [18]. Little is known about the physiological function of the VRK2 gene. It is a serine-threonine which is potentially involved in neural proliferation and migration due to its interaction with multiple biological pathways (i.e., neurite initiation and axon outgrowth) [19]. The lack of VRK2 interferes with synaptic functions, knockout mice for this gene show changes in their social behavior [20]. In genome wide association studies in humans, VRK2 has been consistently associated with psychiatric and neurodegenerative disorders such as schizophrenia, major depressive disorder and genetic generalized epilepsy. The rs2312147 in the intron of this gene has been repeatedly associated with schizophrenia in large human populations of European and Asian individuals [19].

The FANCL gene codes for an ubiquitin ligase, which is found like VRK2 in a region associated with schizophrenia, due to the SNPs rs11682175 and rs75575209 being found in LD, they were identified by the Schizophrenia Working Group of the Psychiatric Genomics Consortium when studying 36,989 cases and 113,075 controls [21]. In the same way in this work we identified the candidate genes VRK2 and FANCL from SNPs rs134895560 and rs110882543 ($r^2$ = 0.82). Some studies have corroborated the relationship of genes identified through studies of

**Table 3. Least square means of genotype effects of SNPs associated with Brahman temperament traits using a mixed model fitted.**

| Region/Gene | SNP | Intron | P-values | | | Genotype | Least square means | | |
| --- | --- | --- | --- | --- | --- | --- | --- | --- | --- |
| | | | EV | PS | TS | | EV | PS | TS |
| Intergenic | rs135340276 | - | 0.0027 | 0.0510 | 0.0015 | CC | 2.09 [a] | 2.62[a] | 2.36[a] |
| | | | | | | TC | 2.89 [b] | 3.15[ab] | 3.02[b] |
| | | | | | | TT | 3.34 [b] | 3.49[b] | 3.41[b] |
| Intergenic | rs134895560 | - | 0.0002 | 0.6969 | 0.0088 | AA | 1.87 [a] | 2.83 | 2.35[a] |
| | | | | | | AG | 2.95 [b] | 3.05 | 3.00[b] |
| | | | | | | GG | 3.36 [b] | 3.12 | 3.23[b] |
| Intergenic | rs110190635 | - | 0.0005 | 0.0679 | 0.0019 | AA | 1.72 [a] | 2.32 | 2.02[a] |
| | | | | | | AG | 2.44 [a] | 3.10 | 2.77[b] |
| | | | | | | GG | 3.23 [b] | 3.03 | 3.13[b] |
| Intergenic | rs42949831 | - | 0.0192 | 0.8468 | 0.3732 | CC | 2.09 [a] | 3.05 | 2.57 |
| | | | | | | TC | 2.58[ab] | 2.93 | 2.76 |
| | | | | | | TT | 3.21 [b] | 2.83 | 3.02 |
| NRXN3 | rs135982573 | 1 | 0.0138 | 0.9508 | 0.1374 | GG | 2.19 [a] | 2.92 | 2.55 |
| | | | | | | AG | 3.12 [b] | 2.97 | 3.04 |
| | | | | | | AA | 3.08 [b] | 3.04 | 3.06 |
| EXOC4 | rs109393235 | 7 | 0.0005 | 0.1687 | 0.0023 | AA | 2.31 [a] | 2.83 | 2.57[a] |
| | | | | | | AG | 3.44 [b] | 3.46 | 3.45[b] |
| | | | | | | GG | 4.95 [b] | 3.32 | 4.14[ab] |
| Intergenic | rs109531929 | - | 0.0082 | 0.1936 | 0.0141 | AA | 3.31 [b] | 3.16 | 3.21[b] |
| | | | | | | AG | 2.76[ab] | 3.15 | 2.95[b] |
| | | | | | | GG | 2.09 [a] | 2.67 | 2.38[a] |
| Intergenic | rs135087545 | - | <0.0001 | 0.2098 | 0.0005 | AA | 2.82[ab] | 3.11 | 2.95[ab] |
| | | | | | | AG | 1.86[b] | 2.69 | 2.27[b] |
| | | | | | | GG | 3.16[a] | 3.16 | 3.15[a] |
| Intergenic | rs41839733 | - | 0.0293 | 0.6348 | 0.3035 | CC | 3.14[b] | 2.77 | 2.95 |
| | | | | | | TC | 2.55[ab] | 3.08 | 2.81 |
| | | | | | | TT | 2.11[a] | 2.90 | 2.50 |
| Intergenic | rs42486577 | - | 0.0014 | 0.0052 | 0.0002 | CC | 2.85[b] | 3.22[b] | 3.03[b] |
| | | | | | | TC | 1.85[a] | 2.26[a] | 2.06[a] |
| | | | | | | TT | - | - | - |
| CACNG4 | rs136661522 | 1 | 0.0023 | 0.2785 | 0.0190 | AA | 2.93[a] | 2.99 | 2.95[a] |
| | | | | | | AG | 2.20[b] | 3.03 | 2.62[ab] |
| | | | | | | GG | 1.23[b] | 2.15 | 1.70[b] |
| Intergenic | rs110882543 | - | 0.0062 | 0.4990 | 0.0609 | AA | 2.14[a] | 2.83 | 2.49 |
| | | | | | | AG | 2.85[b] | 3.14 | 2.99 |
| | | | | | | GG | 3.36[b] | 2.77 | 3.07 |
| SLC9A4 | rs110864071 | 2 | 0.0003 | 0.1005 | 0.0008 | TT | 2.94[a] | 3.17 | 3.05[a] |
| | | | | | | TC | 1.82[b] | 2.61 | 2.21[b] |
| | | | | | | CC | 1.40[b] | 2.33 | 1.91[b] |
| SLC9A4 | rs109722627 | 8 | 0.0002 | 0.3371 | 0.0044 | GG | 3.43[a] | 3.13 | 3.28[a] |
| | | | | | | TG | 2.32[b] | 3.01 | 2.66[b] |
| | | | | | | TT | 1.94[b] | 2.61 | 2.27[b] |

[a,b] Means with different superscript by trait and SNP are significantly different ($P < 0.05$). EV = Exit velocity, PS = Pen Score, TS = Temperament Score.

GWAS with schizophrenia; such is the case of *FANCL* which has been identified by 2 studies of TWAS (Transcriptome-wide association study) as a gene with transcriptome-wide significant association from different samples of patients with schizophrenia [22, 23]. The *SLCO3A1* gene is abundantly expressed in the postnatal brain of mice [24]. This gene participates in networks of genes involved in neurological and developmental disorders [25]. In humans through GWAS, one SNP has been identified within this gene, associated with Parkinson's disease [26]. Interestingly, here we identified two SNPs located at the SLC family at *SLC18A2*, which were associated with temperament traits in the studied population. Garza-Brenner et al. [27] also identified a SNP in this gene with an effect on Pen Score in Charolais cattle.

### Intronic temperament-associated SNPs

Five SNPs located at introns of genes *NRXN3*, *EXOC4*, *CACNG4* and *SLC9A4* in Brahman cattle were identified. Although none of these genes have been previously identified with influence on bovine temperament, we could hypothesize their role on the basis of the reported genomic studies where they have been identified. In humans, these genes have been associated with different neurodegenerative disorders (i.e., Alzheimer and Parkinson's disease) and neuropsychiatric disorders (i.e. autism and schizophrenia); for most of them, genetic variations and polymorphisms have been reported in association with a particular expression of each these diseases. Considering that the aforementioned disorders have as a common denominator, aspects related to the expression of behaviors as a response to stress and fear, and that temperament is a reflection of an animal´s stress response, innate fear or reaction to handling, as well as animal´s aggressiveness [6, 28], all of the genomic information about the genes and pathways previously related in humans could allow us to see patterns and consider the reported pathway interactions as candidates to also explain bovine temperament. Thus for example, the rs135982573 marker identified with significant effect for EV, is located at intron one of the *NRXN3* genes; neurexins (*NRXN*s) act predominantly at the presynaptic terminal in neurons and play essential roles in neurotransmission and differentiation of synapses [29]. In humans, genetic variations in neuroxins have been associated with disorders affecting cognition and behavior via molecular mechanisms including cell adhesion, synaptic transmission and neurotransmitter secretion, [30]. In a bovine population, Qanbari et al. [31] found candidate selection regions harboring genes related to phenotypes relevant to domestication, such as neurobehavioral functioning. *NRXN1* has been proposed as a candidate gene for behavioral traits such as a moderate temperament. Although the molecular basis of the effect of *NRXN3* with temperament in cattle has not been directly determined, according to the STRING database (https://string-db.org/), this gene has interactions with the *AMPA2* (*GRIA2*) gene, supported by text mining and co-expression. Lindholm-Perry et al. [32] reported that a SNP near *GRIA2* was nominally associated with flight speed, a predictive indicator of bovine temperament. This interaction can be supported by the fact that both families of genes Neurexin and *AMPA* receptors are sorted by *SorCS1* receptor, regulating neurexin and *AMPAR* surface trafficking [33]. Further studies are needed to investigate the effect of these interacting genes and their genetic variation influencing bovine temperament.

Association of the SNP rs109393235 located at intron 7 of the *EXOC4* gene, was also found. This gene has been implicated in insulin processing and protein metabolism [34]. It has been associated with reproductive traits (age at first calving) in Canchim beef cattle using a GWAS approach [35]. Interesting, in humans, intronic polymorphisms and other genetic variations have been associated neurological disorders such as Alzheimer's, dementia, schizophrenia and personality disorders [36].

The rs136661522 is located at intron 1 of the *CACNG4* gene. The *CACNG4* gene in cattle has been associated with milk yield traits in Polish Holstein dairy cattle [37] and with sperm motility in cattle, humans and mice [38]. The Voltage-gated L-type calcium channels (VLCC) play an important role both in the nervous system (intracellular flow of $Ca^{2+}$ by action potentials in the synapse) and in the cardiovascular system (regulates excitation-contraction coupling in the heart), because the *CACNG1* to *CACNG8* genes have been identified as regulators of VLCC function (activation and inactivation), it has been shown that *CACNG4* interacts physically with the cardiac voltage-gated $Ca^{2+}$ channel, regulating its function. However; its implications are yet unknown [39]. This is relevant because bovine temperament affects the activation of cardiovascular system. Because this gene is expressed in the human brain, it has been identified as a candidate gene to be involved in the risk of susceptibility to schizophrenia, since SNP near this gene showed a significant effect in a human population [40].

The *SLC9A4* gene, which harbors 2 SNPs in introns 2 and 8, respectively, is part of the *SLC* family, which, as previously mentioned, transports various molecules, such as inorganic ions, fatty acids, saccharides and neurotransmitters [41]. *SLC9A4* corresponds to a group of $Na^+/H^+$ exchangers, which are expressed in the stomach, small intestine, colon, skeletal muscle and brain and it has been associated with metabolic diseases [42]. However, given that in mice it is mostly expressed in the hippocampus of the brain, it could be expected that in some way, it is involved in the expression of temperament [42]. Some genes of the *SCL9A* family (*SLC9A1*) have been associated with neurological disorders, such as epilepsy and bovine genes of the SLC family, such as *SLC18A2* and *SLC6A4*, have been identified as candidates associated with bovine temperament, in Charolais populations, *SLC18A2* has been shown to have a significant effect on Pen Score [27].

At present, the GWAS approach is considered to be a viable strategy to identify disease- and trait-associated genetic variants, interestingly most of the variants discovered (~ 93%) using GWAS, are located in non-coding regions (i.e., introns, promoters, intergenic regions) [43]. In humans, for example, it has been reported that of a total of 920 GWAS studies for different diseases and traits (which included a large number of neurological and behavioural projects), only 4.9% represent genetic variants located in exons, 52.6% in intergenic regions (1Mb of nearest transcriptional start site) and 41% in intronic regions [44]. We used the contrasting group strategy and were able to identify 5 and 9 SNPs in the intronic and intergenic regions associated with EV, respectively. Further investigation is needed to identify candidate genes close to this genomic region, because until now no genes were mapped in this region.

An interesting result was the identification of associated SNP in introns of genes. It is known that the primary sequence of a gene is not limited to expression of a protein, but it also implies elements located outside the protein-coding regions, such as introns, since these may have both direct effects such as alternative splicing, enhanced gene expression or indirect effects, source of new genes, or may harbour several kinds of noncoding functional RNA genes [43]. In humans, neurological disorders such as schizophrenia are under the control of variants that affect gene expression, rather than variants that affect protein structure [21]. Similarly, for this same neurological disorder, it has been identified that there are relationships with intronic SNPs that upregulate the expression of splice variants in the human brain [45].

Further investigation will allow definition of the role of the five intronic SNPs identified in this study; they are potential candidates to address control of bovine temperament, as has been demonstrated for some neurological disorders in humans.

## Conclusions

The genomic analysis of a Brahman population contrasting in their temperaments allowed identification of fourteen intergenic and intragenic SNPs, associated with EV. The

identification of candidate genes harboring or close to these associated SNPs, suggests that cattle temperament may involve clusters of genes implicated in different biological pathways. Further studies are needed to determine how variation in these genes will impact expression of temperament.

## Supporting information

**S1 Dataset. File PLINK MAP.** Contains the BTA, SNP identifier, genetic distance (morgans), base-pair position (bp units) of the SNPs analyzed.
(TXT)

**S2 Dataset. File PLINK PED.** Contains the family ID, individual ID, paternal ID, maternal ID, sex (1 = male; 2 = female), phenotype (-9 missing 0 missing, 1 calm, 2 temperamental) of the animals analyzed.
(TXT)

**S3 Dataset. File temperament measurements.** Contains the animal ID, birth date and values of EV, PS, TS.
(TXT)

## Acknowledgments

The technical assistance of A. W. Lewis and D. A. Neuendorff and the technicians and graduate students in collection of animal data and samples is acknowledged.

## Author Contributions

**Conceptualization:** Ana María Sifuentes-Rincón, Ronald D. Randel.

**Data curation:** Francisco Alejandro Paredes-Sánchez, Ronald D. Randel.

**Formal analysis:** Francisco Alejandro Paredes-Sánchez, Ana María Sifuentes-Rincón, Eduardo Casas, G. Manuel Parra-Bracamonte.

**Funding acquisition:** Ronald D. Randel.

**Investigation:** Francisco Alejandro Paredes-Sánchez, Ana María Sifuentes-Rincón, Ronald D. Randel.

**Methodology:** Francisco Alejandro Paredes-Sánchez, Ana María Sifuentes-Rincón, Williams Arellano-Vera, G. Manuel Parra-Bracamonte, David G. Riley, Thomas H. Welsh, Jr.

**Project administration:** Ana María Sifuentes-Rincón.

**Software:** Francisco Alejandro Paredes-Sánchez.

**Supervision:** Eduardo Casas.

**Writing – original draft:** Francisco Alejandro Paredes-Sánchez, Ana María Sifuentes-Rincón, Eduardo Casas.

**Writing – review & editing:** Francisco Alejandro Paredes-Sánchez, Ana María Sifuentes-Rincón, Eduardo Casas, Williams Arellano-Vera, G. Manuel Parra-Bracamonte, David G. Riley, Thomas H. Welsh, Jr., Ronald D. Randel.

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
