## [Decision Letter · Decision Letter 0]

3 Jun 2020

PONE-D-20-11947

A genome-wide association study identifies novel candidate genes related to cattle temperament

PLOS ONE

Dear Dr. Sifuentes-Rincón,

Thank you for submitting your manuscript to PLOS ONE. After careful consideration, we feel that it has merit but does not fully meet PLOS ONE’s publication criteria as it currently stands. Therefore, we invite you to submit a revised version of the manuscript that addresses the points raised during the review process.

The Materials and Methods section should be revised to include more detail information regarding the analyses performed. Special attention should be given to the organization of this section and manuscript in general. As suggested by one reviewer, the authors should use the most recent genome assembly.

We look forward to receiving your revised manuscript.

Kind regards,

Raluca Mateescu

Academic Editor

PLOS ONE

Journal Requirements:

Additional Editor Comments (if provided):

Reviewers' comments:

Reviewer's Responses to Questions

**Comments to the Author**

1. Is the manuscript technically sound, and do the data support the conclusions?

Reviewer #1: Yes

Reviewer #2: Yes

2. Has the statistical analysis been performed appropriately and rigorously? 

Reviewer #1: Yes

Reviewer #2: No

3. Have the authors made all data underlying the findings in their manuscript fully available?

Reviewer #1: Yes

Reviewer #2: Yes

4. Is the manuscript presented in an intelligible fashion and written in standard English?

Reviewer #1: Yes

Reviewer #2: Yes

5. Review Comments to the Author

Reviewer #1: Majors

The objective of present study was to identify genomic regions and genes associated with cattle temperament. The authors found that fourteen SNPs were associated with exit velocity with allowed the identification of fourteen candidate genes for temperament in Brahman. Also, five intronic SNPs were candidates to address control of bovine temperament. Although there was little problem in the present form, the authors had better revise the following points:

1. As you wrote in L301-304, the aim of this manuscript was to identify the genes associated with temperament, such as stress response, innate fear, reaction to handling and aggressiveness. Considerations seemed to be insufficient in Discussion. You indicated the associations of candidate genes with results in other reports, but these associations were indirect and ambiguous. For example, in L241-248, the readers could not understand the relationship between the SNP and temperament (stress response, innate fear, reaction to handling and aggressiveness) even if “rare mutations and copy-number variations of human neurexin genes have been linked to autism and schizophrenia”. The authors should indicate the mechanism for mutation of neurexin genes induced autism and schizophrenia, and then the relationship between autism/schizophrenia and temperament (stress response, innate fear, reaction to handling and aggressiveness). As for other mutations, you should describe them.

Minor: There were some typographical errors.

L87-8: add the approval number.

L185: insert a space between “0.266),” and “and”

L206-7: indicate not only web address but also references.

L213: add any references.

L245: add any references.

L247: add any references.

L286: add any references.

L288: add any references.

L253: polymorphismos -> polymorphisms

Reviewer #2: A genome-wide association study identifies novel candidate genes related to cattle

Temperament. PONE-D-20-11947

Temperament of animals, a polygenic difficult-to-measure trait, is important in some farming systems. In this study authors presented the results of a GWAS carried out in Brahman cattle. At this point I prefer to focus on major aspects regarding the methodology used.

In particular, statistical analysis and GWAS model are not appropriately described, thus "Materials and Methods" section needs to be improved. Just few examples:

L127-133: how was the genome-wide association study performed? Pedigree size? Model? Software?

L134-143: not clear why you chose these 14 SNP (I can guess why, but it can be not clear for all the readers); Furtherly, it is not clear how you estimated the LSM.

L135 and L145: at this point I do not really understand if you consider all the 3 traits (as in L135) or only EV (as in L145). It seems that you provide results only for EV, but you mention other 2 traits twice in the manuscript (e.g. L95-103).

Apart from this, I also strongly suggest to use the last available and more accurate assembly for Bos Taurus, the ARS-UCD 1.2, for the codification.

Few minor comments:

L61: ‘indirect estimation of heritability’

L64: define the abbreviation ‘SNP’ here

L77: the reader may get confused at this point; I recommend to write ‘in two extreme groups, i.e. ‘calm’ and ‘temperamental’ animals.’

L80: ‘show better growth performance than’

L82: ‘newborn animals’ rather than ‘neonates’

L82-83: ‘would help Brahman farmers in decision making and will improve human safety and animal welfare’

L95: delete reference name

L95-103: a bullet point would be easier to follow; anyway, the list and description provided may fit better in the introduction rather than in M&M, since you use only EV in this study

L108: be consistent with the use of ‘EV’ throughout the text; sometimes I see the full name

L125: rephrase as ‘individuals with more than 10% missing genotypes were also excluded’

L126: provide the % of females and the % of males defined as ‘calm’

6. PLOS authors have the option to publish the peer review history of their article (what does this mean?). If published, this will include your full peer review and any attached files.

Reviewer #1: No

Reviewer #2: Yes: Angela Costa

---

## [Author Response · Author response to Decision Letter 0]

13 Jun 2020

Dear Editor:

Here, you will find our responses to the reviewers’ questions. In general, to complete all of the requested revisions, necessary, major modifications in the methods, results and discussion sections of the manuscript were accomplished. In the former, we reanalyzed the physical position of significant SNPs based on bovine Bos taurus genome (ARS-UCD 1.2) and in the discussion we added information about how previously described genes in the human allowed us to infer their role in bovine temperament. All minor requirements were completed according to reviewer instructions. 

Reviewer #1: Majors

1. As you wrote in L301-304, the aim of this manuscript was to identify the genes associated with temperament, such as stress response, innate fear, reaction to handling and aggressiveness. Considerations seemed to be insufficient in Discussion. You indicated the associations of candidate genes with results in other reports, but these associations were indirect and ambiguous. For example, in L241-248, the readers could not understand the relationship between the SNP and temperament (stress response, innate fear, reaction to handling and aggressiveness) even if “rare mutations and copy-number variations of human neurexin genes have been linked to autism and schizophrenia”. The authors should indicate the mechanism for mutation of neurexin genes induced autism and schizophrenia, and then the relationship between autism/schizophrenia and temperament (stress response, innate fear, reaction to handling and aggressiveness). As for other mutations, you should describe them.

Authors: In L244-L255 of the revised manuscript, we included a more detailed description at the beginning of this discussion in order to introduce to the readers our inferences about how previously described genes in human disorders could be the basis to study their role in bovine temperament. It is worthy to notice that until now all association studies in humans and non-humans are allowing the scientist to explore and hypothesize about the genetic architecture of complex traits. The results of these GWAS investigations are the starting point to study at the molecular level how the discovered candidate genes are involved in expression of the trait. For most of these genes the mutation mechanisms are still unknown, but a wide number of studies support its association to expression of the disorder.

Minor: There were some typographical errors.

L87-8: add the approval number.

L185: insert a space between “0.266),” and “and”

L206-7: indicate not only web address but also references.

L213: add any references.

L245: add any references.

L247: add any references.

L286: add any references.

L288: add any references.

L253: polymorphismos -> polymorphisms

Authors: We attended to all of the listed minor requirements according to reviewer‘s instructions.

Reviewer #2: A genome-wide association study identifies novel candidate genes related to cattle

1. L127-133: how was the genome-wide association study performed? Pedigree size? Model? Software?

Authors: In the L122-L125 of the revised manuscript we include the requested information 

2. L134-143: not clear why you chose these 14 SNP (I can guess why, but it can be not clear for all the readers); Furtherly, it is not clear how you estimated the LSM.

Authors: After a Genome Wide Association Study analysis using a case control approach, these 14 SNPs were significantly associated indicating the putative position of QTLs for our assessed temperament traits (please see L130-131). 

Least Square Means were estimated from the mixed model fitted for each marker included in the model, a complete description was included (please see L144-146).

3. L135 and L145: at this point I do not really understand if you consider all the 3 traits (as in L135) or only EV (as in L145). It seems that you provide results only for EV, but you mention other 2 traits twice in the manuscript (e.g. L95-103).

Authors: We achieved the GWAS analysis with PLINK 1.9 software as cases and controls (please see L122-125); we classified the population into calm and temperamental groups from exit velocity (EV; please see L106-L118), although PLINK already shows that there is a significant association between the significant SNPs and bovine temperament, we also included an additional association analysis (as is described on L138-146) to estimate the quantitative effect of these markers on exit velocity, and other temperament parameters that were available for this population (i.e., Pen Score and Temperament Score).

4. Apart from this, I also strongly suggest to use the last available and more accurate assembly for Bos taurus, the ARS-UCD 1.2, for the codification.

Authors: As is described in L131-L134, we reanalyzed the physical position of significant SNPs based on the bovine Bos taurus genome (ARS-UCD 1.2), because of this update additional precision in the number of the described genes were needed in the results section and in some tables. All changes were highlighted.

Few minor comments:

L61: ‘indirect estimation of heritability’

L64: define the abbreviation ‘SNP’ here

L77: the reader may get confused at this point; I recommend to write ‘in two extreme groups, i.e. ‘calm’ and ‘temperamental’ animals.’

L80: ‘show better growth performance than’

L82: ‘newborn animals’ rather than ‘neonates’

L82-83: ‘would help Brahman farmers in decision making and will improve human safety and animal welfare’

L95: delete reference name

L95-103: a bullet point would be easier to follow; anyway, the list and description provided may fit better in the introduction rather than in M&M, since you use only EV in this study

L108: be consistent with the use of ‘EV’ throughout the text; sometimes I see the full name

L125: rephrase as ‘individuals with more than 10% missing genotypes were also excluded’

L126: provide the % of females and the % of males defined as ‘calm’

Authors: We attended to all of the listed minor requirements according to reviewer instructions

---

## [Decision Letter · Decision Letter 1]

9 Jul 2020

PONE-D-20-11947R1

A genome-wide association study identifies novel candidate genes related to cattle temperament

PLOS ONE

Dear Dr. Sifuentes-Rincón,

Thank you for submitting your manuscript to PLOS ONE. After careful consideration, we feel that it has merit but does not fully meet PLOS ONE’s publication criteria as it currently stands. Therefore, we invite you to submit a revised version of the manuscript that addresses the points raised during the review process.

Please revise the manuscript in response to Reviewer 2 who suggested a couple of small editorial changes.

We look forward to receiving your revised manuscript.

Kind regards,

Raluca Mateescu

Academic Editor

PLOS ONE

Reviewers' comments:

Reviewer's Responses to Questions

**Comments to the Author**

1. If the authors have adequately addressed your comments raised in a previous round of review and you feel that this manuscript is now acceptable for publication, you may indicate that here to bypass the “Comments to the Author” section, enter your conflict of interest statement in the “Confidential to Editor” section, and submit your "Accept" recommendation.

Reviewer #1: All comments have been addressed

Reviewer #2: All comments have been addressed

2. Is the manuscript technically sound, and do the data support the conclusions?

Reviewer #1: Yes

Reviewer #2: Yes

3. Has the statistical analysis been performed appropriately and rigorously? 

Reviewer #1: Yes

Reviewer #2: Yes

4. Have the authors made all data underlying the findings in their manuscript fully available?

Reviewer #1: Yes

Reviewer #2: Yes

5. Is the manuscript presented in an intelligible fashion and written in standard English?

Reviewer #1: Yes

Reviewer #2: Yes

6. Review Comments to the Author

Reviewer #1: (No Response)

Reviewer #2: Overall, authors answered to my questions and my major concern was solved.

I am satisfied with the new version.

Some minor changes and suggestions

• Table 3: Least square means would be easier to read if you add a row between different SNP in order to separate them. In addition, you can remove the SE, as you already have the letters of the t-test. Note that usually superscript letters are adopted for this purpose.

• For consistency, consider to use always ‘BTA’ instead of ‘Chr’ or ‘chromosome’ throughout the whole manuscript (see line 69, 71, 72, 74, 150, 204, 205, 206…)

• Line 133: ‘on the last available Bos Taurus genome’

• Lines 142-143: I recommend to briefly explain why you used year of birth as random effect

• Table 1: position in Mb is sufficient

• Table 2: ‘gene ID’ is not needed

• Should sub-titles be smaller than the main section title? (see for example “Discussion” and Line 202 or Line 242)

• Consider to change the title as ‘Novel genes involved in the genetic architecture of temperament in Brahman cattle’ …I think it sounds more appealing

• It makes sense to add the (phenotypic and genetic) correlations available in the literature for cattle exit velocity, pen score, and temperament score

Best regards

7. PLOS authors have the option to publish the peer review history of their article (what does this mean?). If published, this will include your full peer review and any attached files.

Reviewer #1: No

Reviewer #2: **Yes: **Angela Costa

---

## [Author Response · Author response to Decision Letter 1]

16 Jul 2020

Dear Editor:

Here, you will find our responses to Reviewer #2 changes and suggestions. 

In general, minor changes in format were necessary and were attended as is detailed. The major review was a change in the manuscript title, however, we are agreeing with the Reviewer ´s suggestion and accepted the new title.

Reviewer #2: Overall, authors answered to my questions and my major concern was solved. I am satisfied with the new version.

Some minor changes and suggestions

• Table 3: Least square means would be easier to read if you add a row between different SNP in order to separate them. In addition, you can remove the SE, as you already have the letters of the t-test. Note that usually superscript letters are adopted for this purpose.

Authors: We attended the requirements according to reviewer recommendation.

• For consistency, consider to use always ‘BTA’ instead of ‘Chr’ or ‘chromosome’ throughout the whole manuscript (see line 69, 71, 72, 74, 150, 204, 205, 206…)

Authors: We attended the requirements according to reviewer recommendation (see Table 1 and lines 148, 151, 162, 168, 169, 171, 172, 202-204, 206, 207, 470).

• Line 133: ‘on the last available Bos Taurus genome’

Authors: We attended the requirements according to reviewer recommendation (see line 130).

• Lines 142-143: I recommend to briefly explain why you used year of birth as random effect

Authors: A brief justification was included. Please see L139-142.

• Table 1: position in Mb is sufficient

Authors: We attended the requirements according to reviewer recommendation.

• Table 2: ‘gene ID’ is not needed

Authors: We attended the requirements according to reviewer recommendation.

• Should sub-titles be smaller than the main section title? (see for example “Discussion” and Line 202 or Line 242)

Authors: We attended the requirements according to reviewer recommendation and change some subtitles. Please see lines 116, 179, 200, 240. 

• Consider to change the title as ‘Novel genes involved in the genetic architecture of temperament in Brahman cattle’ …I think it sounds more appealing

Authors: We appreciate the suggestion and agree with the proposed title, it was changed according to reviewer recommendation

• It makes sense to add the (phenotypic and genetic) correlations available in the literature for cattle exit velocity, pen score, and temperament score

Authors: Required information was included. Please see L105-106.

---

## [Editor Report · Decision Letter 2]

4 Aug 2020

Novel genes involved in the genetic architecture of temperament in Brahman cattle

PONE-D-20-11947R2

Dear Dr. Sifuentes-Rincón,

We’re pleased to inform you that your manuscript has been judged scientifically suitable for publication and will be formally accepted for publication once it meets all outstanding technical requirements.

Kind regards,

Raluca Mateescu

Academic Editor

PLOS ONE
---

## [Editor Report · Acceptance letter]

11 Aug 2020

PONE-D-20-11947R2 

Novel genes involved in the genetic architecture of temperament in Brahman cattle 

Dear Dr. Sifuentes-Rincón:

I'm pleased to inform you that your manuscript has been deemed suitable for publication in PLOS ONE. Congratulations! Your manuscript is now with our production department. 

Kind regards, 

on behalf of

Dr. Raluca Mateescu 

Academic Editor

PLOS ONE